# Bears into the Niche-Space: Phylogeography and Phyloclimatic Model of the Family Ursidae

Carlos Luna-Aranguré [1,2,*] and Ella Vázquez-Domínguez [3,4,*]

1 Instituto de Ciencias de la Atmósfera y Cambio Climático, Universidad Nacional Autónoma de México, Ciudad de México 04510, Mexico
2 Posgrado en Ciencias Biológicas, Universidad Nacional Autónoma de México, Edificio D, 1er Piso, Unidad de Posgrado, Ciudad de México 04510, Mexico
3 Departamento de Ecología de la Biodiversidad, Instituto de Ecología, Universidad Nacional Autónoma de México, Ciudad de México 04510, Mexico
4 Conservation and Evolutionary Genetics Group, Estación Biológica de Doñana (EBD-CSIC), 41092 Sevilla, Spain
* Correspondence: carlos.luna@pincc.unam.mx (C.L.-A.); evazquez@ecologia.unam.mx (E.V.-D.)

**Abstract:** Assessing niche evolution remains an open question and an actively developing area of study. The family Ursidae consists of eight extant species for which, despite being the most studied family of carnivores, little is known about the influence of climate on their evolutionary history and diversification. We evaluated their evolutionary patterns based on a combined phylogeography and niche modeling approach. We used complete mitogenomes, estimated divergence times, generated ecological niche models and applied a phyloclimatic model to determine the species evolutionary and diversification patterns associated with their respective environmental niches. We inferred the family evolutionary path along the environmental conditions of maximum temperature and minimum precipitation, from around 20 million years ago to the present. Our findings show that the phyloclimatic niches of the bear species occupy most of the environmental space available on the planet, except for the most extreme warm conditions, in accordance with the wide geographic distribution of Ursidae. Moreover, some species exhibit broader environmental niches than others, and in some cases, they explore precipitation axes more extensively than temperature axes or vice versa, suggesting that not all species are equally adaptable to these variables. We were able to elucidate potential patterns of niche conservatism and evolution, as well as niche overlapping, suggesting interspecific competitive exclusion between some of the bear species. We present valuable insights into the ecological and evolutionary processes driving the diversification and distribution of the Ursidae. Our approach also provides essential information for guiding effective conservation strategies, particularly in terms of distribution limits in the face of climate change.

**Keywords:** bears; niche modeling; NicheRover; phyloclimatespace; phylogeography; Ursidae

## 1. Introduction

A crucial question in evolutionary biology is understanding how the environment can shape the diversification patterns and geographic distribution of species [1,2]. Phylogeography is a discipline that integrates information on the genetic structure of natural populations, their geographic distribution, and genealogy [3]. Ecological niche modeling is a set of techniques and specialized tools for studying the response of living beings to climatic variables that constitute their environment and their associated geographic distribution [4,5]. Recent evidence shows that the combined study of these two disciplines helps assessing the influence of environmental factors on the distribution of genetic variability throughout geographical ranges and the role those factors play in the divergence of species across their distribution and landscapes [1,6–8]. The synergy of these two disciplines also

helps disentangling the role of ecological and environmental factors in the divergence of lineages within a taxon [9].

The study of the ecological niche in terms of species evolutionary patterns is based on two main predictions. One is niche conservatism that refers to the shared ecological characteristics of species or lineages with their ancestors, namely, that species tend to share with their ancestors the biological and physiological characteristics that allow them to adapt to their environment [2,10,11]. Alternatively, niche evolution denotes the adaptation and persistence of populations or lineages to novel environmental conditions [12,13]. It is noteworthy that although niche modeling studies attempt to discern between these two hypotheses, it has rarely been explicitly determined how the niches of species evolve [13–15]. Therefore, assessing niche evolution remains an open question and an actively developing area of study.

Bears are a taxon of great biological interest and intensively studied. The bear family Ursidae has a complex evolutionary history despite being evolutionarily young, with origins dating back to the Late Oligocene and the Early Miocene [16]. The family underwent rapid radiation between the mid-Pliocene and the Early Pleistocene, giving rise to most of the known bear species [17–19]. Although there is evidence of approximately 100 extinct species in the fossil record, in the present, 8 bear species are recognized: the American black bear (*Ursus americanus*), the brown bear (*Ursus arctos*), the polar bear (*Ursus maritimus*), the Asiatic black bear (*Ursus thibetanu*s), the giant panda (*Ailuropoda melanoleuca*), the spectacled bear (*Tremarctos ornatus*), the sun bear (*Helarctos malayanus*), and the sloth bear (*Melursus ursinus*).

Notably, despite ample available information about the taxonomy, distribution, abundance, ecology, and paleontology of the family, little is actually known about the historical and evolutionary processes that define the evolutionary history, genealogy, and current geographic distribution of bears [8]. Understanding the relationship between patterns of diversification and niche divergence in the family Ursidae remains a challenge in biogeography and evolutionary biology [2]. Hence, bears represent an ideal group for evaluating evolutionary patterns based on a combined phylogeography and niche modeling approach.

In this study, we aimed to describe the patterns of the environmental niche evolution of the living Ursidae species. To this end, we implement the theoretical and methodological tools of ecological niche modeling and phylogeography to model their evolutionary trajectories and diversification using mitochondrial genome data. Our study is based on two premises: First, that analyzing niche relationships among closely related species, particularly those that have evolved recently (i.e., in the last 20 million years), enables assessing their ecological differentiation and the mechanisms responsible for their diversification [1,20]. Second, that precipitation and temperature are abiotic factors with close relationships to the Earth's climates and biomes and hence with the organisms that inhabit those realms [21]. These climatic variables can be useful surrogates of the species biogeographic patterns, which are commonly applied for characterizing the environmental niche of species via ecological niche modeling and hypervolume analyses [22].

We followed an integrative approach that encompasses the following: (i) compile a comprehensive set of mitogenome sequences and geographic distribution records of the eight species of the Ursidae family; (ii) analyze the molecular information within a phylogeographic context; (iii) generate ecological niche models capable of characterizing the environmental conditions that define the habitat of each of the eight species; and (iv) apply a phyloclimatic model to determine the species evolutionary and diversification patterns associated with their respective environmental niche. We predicted that we would be able to reconstruct the historical environmental trajectories of the eight extant bear species and their ancestors, by incorporating the environmental centroid and ellipsoid niche of the species which were based on the most extreme conditions of precipitation and temperature. Thereby, we would more accurately define the climatic conditions inhabited by each species in the present, providing insights to unravel the environmental basis of their distinct geographic distribution. The results obtained contribute both to a better

understanding of the evolutionary patterns of a taxonomic group of great interest such as bears and to the interdisciplinary integration of niche modeling and phylogeography. Such integration will undoubtedly enable a more comprehensive study of the evolution of the ecological niche of species and their evolutionary lineages [1,7,8,13]. This information is extremely valuable to anticipate potential risks associated with global climate change in the future [23,24].

## 2. Materials and Methods

### 2.1. Occurrence Records

We built a database of occurrences with all the bear species' records found in the GBIF (http://gbif.org (accessed on 14 February 2023)) and the scientific literature, filtering all records outside the present and historical ranges reported by the International Union for the Conservation of Nature [25]. Duplicated and misplaced data were eliminated manually, while also avoiding clusters of occurrences to reduce the density of oversampled areas. Additionally, we used the spThin Package [26] in R [27] to eliminate the problems associated with biases in spatial sampling during the niche modeling process. The software's protocol we followed includes distance-based thinning, where points are selected to maximize the distance from each other or by clustering techniques; consideration of spatial autocorrelation to ensure thinned datasets retain similar spatial autocorrelation structures; utilization of spatial interpolation techniques to estimate distributions and select representative points; density-based methods to preserve density distributions across the study area; and iterative algorithms that iteratively refine the selection of points to better match the spatial characteristics of the original datasets [26]. We chose a thinning distance equal to the pixel size of the environmental layers utilized in the study (10 arc-minutes) to keep at least one occurrence per pixel. Our final dataset included 126 records of *Ailuropoda melanoleuca*, 2174 of *Tremarctos ornatus*, 71 of *Melursus ursinus*, 35 of *Helarctos malayanus*, 194 of *Ursus thibetanus*, 756 of *Ursus americanus*, 9107 of *Ursus arctos*, and 559 of *Ursus maritimus* (Figure 1).

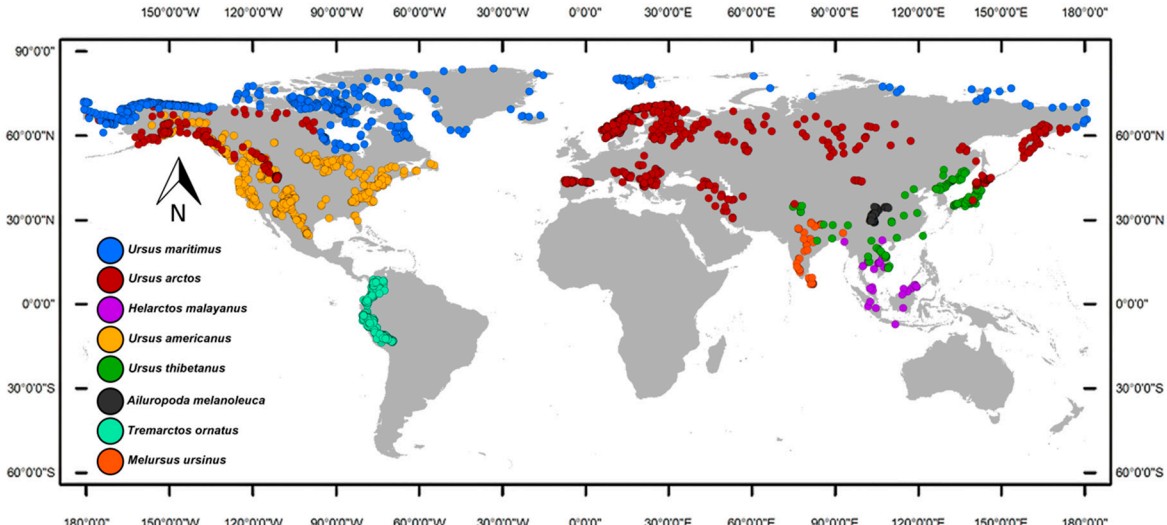

**Figure 1.** Geographic distribution of the family Ursidae occurrences, obtained from the scientific literature and global databases. The filtered database includes 126 records of *Ailuropoda melanoleuca*, 2174 of *Tremarctos ornatus*, 71 of *Melursus ursinus*, 35 of *Helarctos malayanus*, 194 of *Ursus thibetanus*, 756 of *Ursus americanus*, 9107 of *Ursus arctos*, and 559 of *Ursus maritimus*.

### 2.2. Bioclimatic Variables

We obtained the 19 bioclimatic variables from WorldClim [28] with a resolution of 10 arc-minutes (approximately 18 km at the Equator). These variables are derived from interpolation of observed climate data from meteorological stations worldwide and sub-

jected to rigorous modeling and validation processes to assess their accuracy and reliability. Next, we used the NicheToolBox package [29] in R to identify and filter autocorrelated variables to avoid collinearity among predictors, avoid biased parameter estimates, enhance predictive performance and meet assumptions of statistical tests [30]. After this process, six variables were selected and used for further analyses, including annual mean temperature, maximum temperature of the warmest month, minimum temperature of the coldest month, annual mean precipitation, precipitation of the wettest quarter, and precipitation of the driest quarter. Different combinations of these variables play critical roles in shaping the geographic ranges of species by reflecting the species' physiological limits and range boundaries [31,32].

### 2.3. Environmental Characterization

To characterize the environmental niche of the eight bear species, we extracted the environmental values from all their filtered occurrences and used the NicheROVER package [33] in R to analyze the data, generate environmental niche models and quantify their probability of overlap. NicheRover is based on a probabilistic method intended for multi-dimensional niche indicators (e.g., environmental variables); it considers the uncertainty of the data by applying a Bayesian framework and where the number of iterations is defined by the user. It generates ellipsoid models (niche regions) based on n-dimensional continuous data. The ellipsoid models enable the user to quantify and visually assess niche overlapping patterns and niche geometry. Importantly, ellipsoid models yield directional probabilities of the overlapping patterns. In addition, this method takes into account the variation of the number of occurrences (i.e., species with fewer occurrences and others with more) prior to generating overlap estimates between species. Hence, it provides significantly more robust results for niche characterization compared to the traditional percentages of overlap and to those that do not consider the uncertainty associated with the data [33–35]. We defined the niche regions for each species with the ellipsoidal models, and we also obtained their respective centroids using the package dplyr [36] in R, which is simply the arithmetic mean of all the points in the set, by calculating the average of the X and Y coordinates of all the points.

### 2.4. Time-Calibrated Bayesian Ultrametric Tree

To obtain an ultrametric tree with diversification times to be used with the phyloclimatic analysis (see Section 2.5), we performed a three-step procedure. First, we performed a comprehensive search in GenBank (https://www.ncbi.nlm.nih.gov (accessed on 22 Novemeber 2022)) and the scientific literature to gather the most complete mitogenome sequence information for each species. We selected sequences based on length (base pairs) which was key for our objective, to obtain a robust ultrametric tree, and that jointly had the lowest missing data as well as higher depth and coverage. Finally, it was crucial that they were available for the eight species (e.g., brown bears have ample genomic information, while for the sloth bear there were only 5 sequences available). Our final selection included eight sequences provided by [37] in GenBank. The mitogenome sequences of approximately 15,670 bp long were aligned using Geneious v.7.1.9 [38], with a cost matrix of a 93% similarity (Supplementary Materials: sequence alignment). Secondly, GTR was selected as the best substitution model with jModelTest 2.0 [39] and PhyML [40], using the option of smart model selection for the latter [41]. Finally, we estimated the time to the most recent common ancestor with a Bayesian Markov chain Monte Carlo (MCMC) approach, with priors of an uncorrelated lognormal relaxed molecular clock, a constant population size, and the GTR model. A mitogenome sequence of the red panda (*Ailurus fulgens*) was used as an outgroup [42]. We provided seven calibration points based on [43] and obtained a final ultrametric tree by sampling every 1000 iterations for 107 generations and 10% burn-in. Convergence and stationarity were confirmed with Tracer v.1.6 [44].

*2.5. Phyloclimatic Analysis*

With the objective to discern the phylogeographic patterns along each species' environmental space, we applied the phyloclimatespace approach, originally proposed by [45]. This approach is a method for exploring the relationship between phylogeny (evolutionary history) and climate space (the multidimensional space defined by climatic variables). It enables us to visualize evolution through climate space and to evaluate both the environmental niche and the phylogenetic diversification patterns of the studied species. The model was performed with phytools [46] in R, which uses two datasets: (1) a time-calibrated Bayesian ultrametric tree for ancestral reconstruction; (2) the x and y environmental values of every tip on the tree (in this case the environmental centroid for each species), to reconstruct the environmental states of the ancestors. The maximum temperature and the minimum precipitation were the selected variables, because they adequately condense information about the tolerance limits of the species [47] and maximize the differences in their environmental niches. Phytools reconstructs the environmental ancestral states (internal nodes) of the lineages using a restricted maximum likelihood (REML) ancestral state reconstruction [48], assuming a Brownian model of evolution. The tips (terminal nodes) and internal nodes are plotted on the climate space. Importantly, the reconstructed nodes preserve the phylogenetic relationships in strict agreement with the topology and branch lengths of the phylogenetic tree of the species [45].

**3. Results**

*3.1. Niche Modeling Analysis*

NicheROVER results showed ellipsoid models useful for probabilistic quantification of the niches and their overlapping patterns (Table 1; Figure 2) and for the visualization of the niche geometry patterns of the family Ursidae.

**Table 1.** Estimates of ellipsoid niche overlap probability (%) obtained with NicheRover and an alpha value of 0.99. The values are directional, and they represent the probability of overlap of the niche of one species (rows) within the niche of another species (columns). The six variables included in the analysis were mean, maximum, and minimum values of temperature and precipitation. Higher overlap values are indicated in bold.

|  | *A. mel* | *T. orn* | *M. urs* | *H. mal* | *U. thi* | *U. ame* | *U. arc* | *U. mar* |
|---|---|---|---|---|---|---|---|---|
| *A. mel* |  | 1.30 | 0.38 | 0 | **93.22** | **46.85** | 0 | 0 |
| *T. orn* | 0 |  | 9.02 | 16.56 | 9.62 | 23.31 | 0 | 0 |
| *M. urs* | 0 | 2.20 |  | 15.22 | 28.02 | 0.60 | 0 | 0 |
| *H. mal* | 0 | 15.15 | 22.89 |  | 30.96 | 0.79 | 0 | 0 |
| *U. thi* | 0.04 | 1.43 | 5.72 | 2.06 |  | **39.13** | 0.11 | 0.03 |
| *U. ame* | 0.01 | 0.29 | 0.15 | 0.02 | **49.05** |  | 7.16 | 0.45 |
| *U. arc* | 0 | 0.01 | 0 | 0 | 19.32 | **98.66** |  | 8.22 |
| *U. mar* | 0 | 0.02 | 0 | 0 | 5.20 | **66.19** | 7.66 |  |

The niche of the giant panda exhibited one of the smallest sizes with respect to the rest of the family; however, it is the one that collectively showed the highest percentage of environmental overlap, exhibiting 93.2% with the Asiatic black bear and 46.8% with the black bear. The niche of the spectacled bear had its greatest environmental overlap with those of the American black bear (23.3%) and the sun bear (16.6%). The niche of the sloth bear showed the lowest percentage of overlap with the rest of the members of the family, with 28.02% with the Asian black bear and 15.2% with the sun bear. This species also exhibited the largest range of precipitation conditions and the smallest with respect to temperature. The niche of the sun bear showed the highest values in temperature conditions and presented its highest percentage of overlap with the Asian black bear (30.96%), the sloth bear (22.89%), and spectacled bear (15.15%) (Table 1; Figure 2).

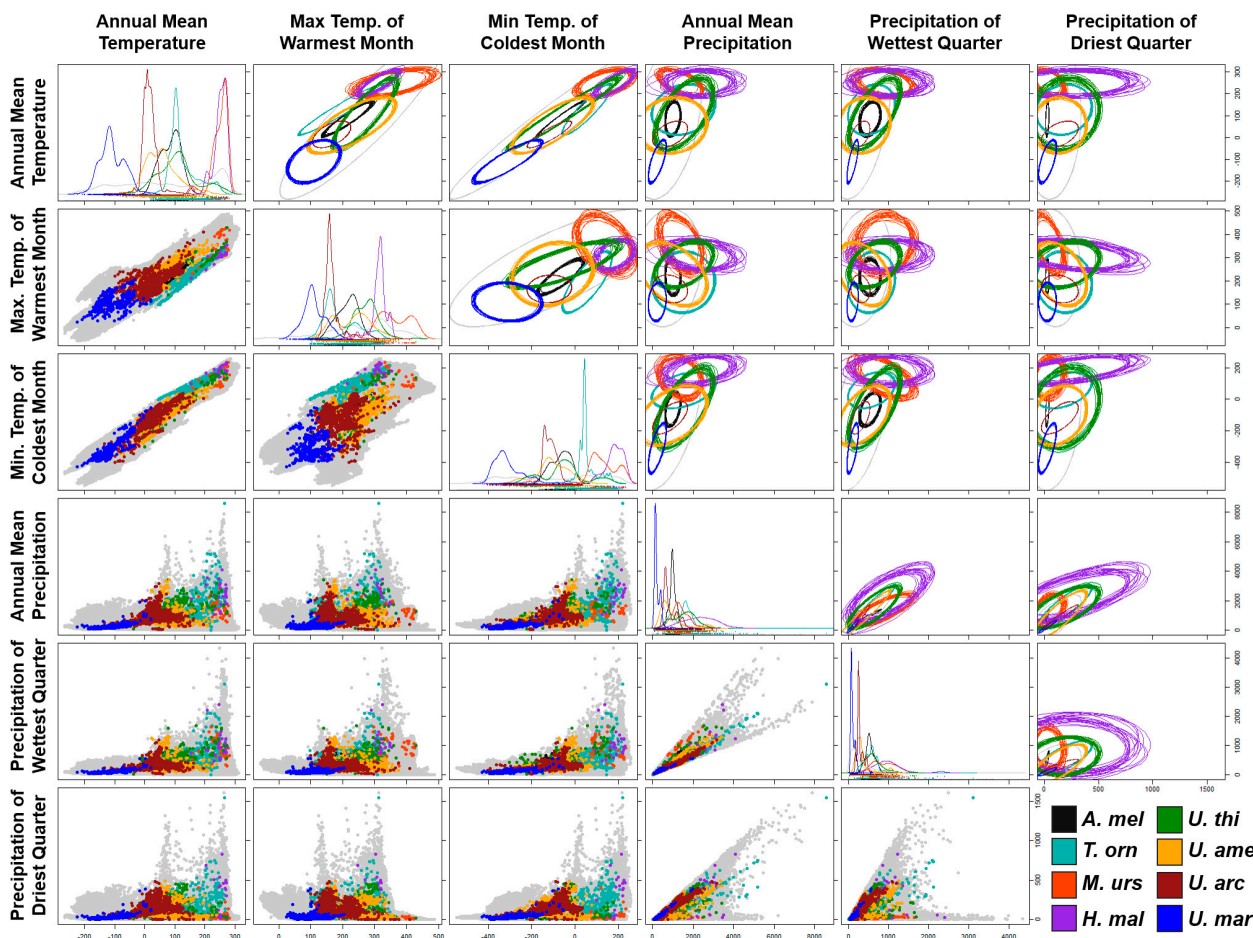

**Figure 2.** Ecological niche models based on isotopic methods, showing one-dimensional density plots, two-dimensional scatterplots, and ellipsoids for each species of the family Ursidae. Ellipsoids per species show 20 iterations of estimated niche amplitude and niche overlapping patterns. The temperature values were multiplied by 10 and shown on a Celsius scale. Precipitation values are shown in millimeters of rainfall. The grey color represents the environmental values of 100,000 equidistant random land points from the planet.

The niches of the closely related *Ursus* species, the American black and the Asian black bears, showed considerable breadth in the central regions of each combination of variables, mutually presenting their highest percentages of overlap (49.05% and 39.13% reciprocally); in addition, both presented less than 10% of overlap with the rest of the family members. Despite the wide distribution of the brown bear across the entire Northern Hemisphere, this species had a considerably small niche with respect to its most ancestral relatives. Additionally, this species showed a 98.66% overlap with the American black bear, the highest percentage of overlap of the family that shares distribution in North America. Finally, the polar bear's niche presented the coldest and driest conditions of the entire family, overlapping more with the American black bear (66.19%) than with its closest relative, the brown bear (7.66%) (Table 1; Figure 2).

### 3.2. Phyloclimatic Analysis

The topology of our Bayesian tree based on mitogenome sequences (Figure 3) agrees with previous phylogenies proposed for the family Ursidae [43,49,50]. The giant panda and the spectacled bear are the species that diverged the earliest, ~20.0 and 13.7 million years ago (My), respectively. The other six species diverged much more recently (~5 My) and, notably, two of the genus *Ursus*, the brown and the polar bears, diverged less than 1 My ago.

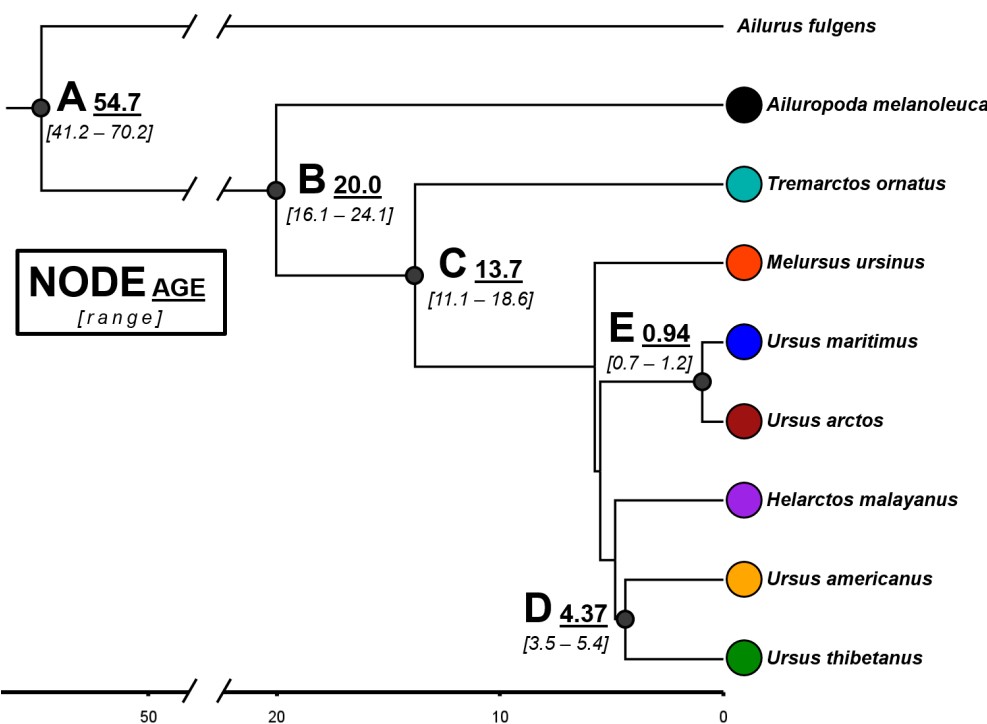

**Figure 3.** Time-calibrated Bayesian tree in million years, based on complete mitochondrial genomes (15,670 bp) of the eight extant species of the family Ursidae. The red panda *Ailurus fulgens* was used as an outgroup. The estimated median time and range are indicated for some nodes. All the indicated nodes have a 95% highest posterior density (HPD) equal to one.

The phyloclimatic results showed a general exclusion pattern among most of the eight bear species (Figure 4). The estimated root was reconstructed between 20 and 25 °C on the axis of the maximum temperature of the warmest month of the year and close to 100 mm of minimum precipitation of the driest month of the year. The two most ancestral lineages of the family fell between 15 and 25 °C, where the spectacled bear exhibited a broader phyloclimatic niche compared to the one estimated for the giant panda. Regarding the sun and the sloth bears, the centroids of their niches occurred in higher temperatures between 30 and 40 °C, that is, they were the two lineages distributed in the warmest conditions. Notably, the branch of the sloth bear extended opposite (along the temperature axis) to that of the most recently diverged species, the polar bear. Contrastingly, the lineage of the sun bear encompassed a wide precipitation (ca. 50–350 mm) but a particularly narrow temperature range (ca. 30–33 °C).

Among the four *Ursus* species, the results showed that the phyloclimate niche of the American black bear lied closest to the reconstructed root of the tree, located near the centroid of the entire family (Figure 4). This lineage had little overlap with its sister species, the Asian black bear, restricted to near 26 °C and 150 mm. There was almost null overlap between the phyloclimate niches of the two lineages that share distribution in southern Asia, the Asian black bear and the sun bear. The smaller phyloclimate niches were represented by three lineages, the brown bear, the polar bear, and the giant panda. Notably, the environmental space near the reconstructed node between the brown and the polar bears did not overlap with the niche of any other living ursid. In particular, the estimated phyloclimate niche of the polar bear included the coldest (9–12 °C) and driest (0–50 mm) conditions of the entire family.

Finally, the phyloclimatic niche of the spectacled bear exhibited a marked environmental overlap with the American black bear and the brown bear (Figure 4), despite their significantly separated geographical distributions: along the Andes Mountain range, in North America, and in practically the entire Northern Hemisphere.

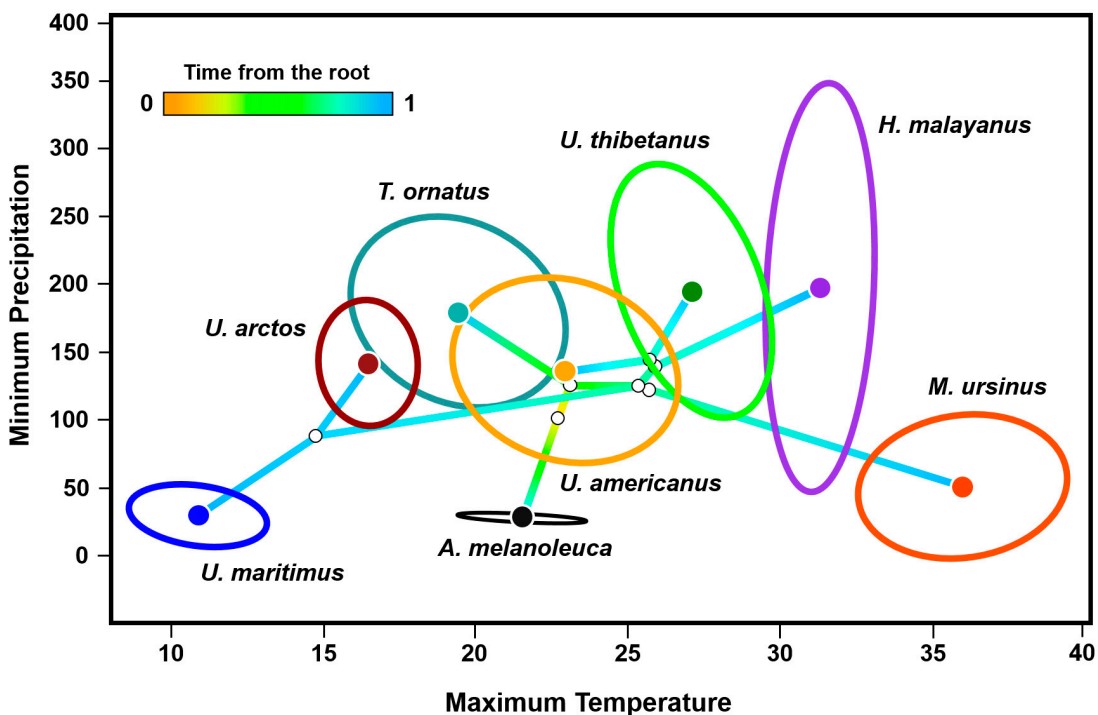

**Figure 4.** Phyloclimatespace results for the eight extant species of the family Ursidae. The model retains the relationships between the complete mitochondrial sequences of the species (Figure 2), positioning each tip of the tree on the combination of precipitation and temperature of its environmental centroid. It shows linear reconstructions of the ecological trajectories of the ancestors (i.e., the internal nodes of the tree and the root). The niche centroid per species was determined from the environmental data extracted from the occurrence records available in GBIF and the scientific literature, while the ellipsoids were estimated by using 95% of the occurrences.

## 4. Discussion

### 4.1. Ecological Niche and Phylogeography of the Extant Bear Species

Variation in climate is closely related to patterns of ecological niche evolution and species diversification [15,51]. Therefore, it is crucial to characterize abiotic ecological niches drawing from phylogeography and evolutionary information [52]. With our phyloclimatic approach, we were able to reconstruct the historical environmental trajectories of the eight extant bear species, based on a combined phylogeographical and ecological niche modeling strategy. Moreover, the estimated environmental trajectories of the bear species and their reconstructed ancestors enabled us to infer the family evolutionary path along the environmental conditions of maximum temperature and minimum precipitation, from around 20 million years ago to the present. Our findings show that the phyloclimatic niches of the bear species occupy most of the environmental space available on the planet, except for the most extreme warm conditions, in accordance with the wide geographic distribution of Ursidae.

The model consistently showed differential use of environmental space and ecological specialization, suggesting interspecific competitive exclusion between some of the bear species (Figure 2). Moreover, some species exhibited broader environmental niches than others and, in some cases, they explored precipitation axes more extensively than temperature axes or vice versa, suggesting that not all species are equally adaptable to these variables. This also highlights the importance of enriching ecological niche projections with extreme values instead of using only average values. We show that our approach improves the categorization of ecological niches both of the species and of their evolutionary lineages, by reconstructing the environmental conditions that they may have inhabited in the past (Figure 3) [6].

### 4.2. Environmental Adaptation, Niche Conservatism, and Niche Evolution

Niche conservatism, a fundamental concept in evolutionary ecology, provides a lens through which we can understand the interplay between organisms and their environments over time. It posits that species tend to retain ancestral ecological characteristics, adapting to environmental changes within the constraints of their inherited niche [2,10,11,53]. Our study offers valuable insights into the dynamics of niche conservatism and evolution within the Ursidae family.

Our findings reveal instances of both niche conservatism and niche evolution across the eight extant bear species. The phyloclimatic niches of the American black bear (*Ursus americanus*) and the Asian black bear (*Ursus thibetanus*), despite their geographic separation and substantial divergence time (ca. 4.37 million years), exhibit a considerable overlap. This suggests a degree of niche conservatism, where both species have retained similar ecological preferences despite millions of years of independent evolution [18,19,54]. However, it is noteworthy that the Asian black bear, the more ancient lineage, is better adapted to a wider variety of wet and warm climatic conditions [55]. This can be associated with asymmetry in ecological niche conservatism, that is, ancient tropical lineages are likely constrained to inhabit warm climates, while groups that evolved more recently and are adapted to tolerate cold temperatures are potentially more flexible, likely due to reduced or less complex biotic interactions [47,55–57].

Conversely, species like the polar bear (*Ursus maritimus*) exemplify niche specialization and evolution towards extreme environmental conditions. Its phyloclimatic niche is markedly constrained to the coldest and driest conditions available on the Northern Hemisphere, reflecting a highly specialized adaptation to its Arctic habitat. This specialization likely evolved in response to selective pressures associated with the polar environment, including dependence on sea ice for hunting and unique physiological adaptations to cold climates, as well as biological features like hibernation and specialized diet [54,58]. It is the only land carnivore in the Arctic Circle that hunts seals and other sea mammals [58]. Notably, its niche is the second smallest followed by the giant panda (*Ailuropoda melanoleuca*). The giant panda is similarly characterized by a rather specialized diet, consisting of a variety of bamboo species. Such restriction to bamboo-dominated habitats has shaped the distribution and evolutionary trajectory of the giant panda [59,60] It is important to highlight that we used modern occurrence (i.e., not fossil) records to generate the ellipsoidal models. Thus, the fact that the present distribution of some species like the giant panda represents only a fraction of what it occupied in the past [59,61,62], suggests that some of the estimated historical niches could be broader.

The relationship between the polar bear and the brown bear (*Ursus arctos*) offers a compelling example of niche conservatism and divergence within the Ursidae family. Despite sharing a common ancestry and significant genetic similarity, these two species have undergone remarkable ecological specialization in response to divergent environmental pressures. While the brown bear maintains a broad environmental niche across its range, encompassing diverse habitats from temperate forests to tundra regions, the polar bear has evolved a highly specialized niche adapted exclusively to the Arctic marine environment. As previously mentioned, this specialization is evident in the polar bear's dependence on sea ice for hunting seals and its physiological adaptations to extreme cold temperatures [58]. Interestingly, despite their ecological divergence, genetic studies have revealed a close evolutionary relationship between polar bears and certain populations of brown bears, particularly those inhabiting coastal regions of Alaska and the Russian Far East [63]. This genetic admixture suggests historical introgression and gene flow between the two species. However, despite these interactions, the polar bear's niche remains distinct, emphasizing the role of ecological specialization in driving species divergence and adaptation. On the other hand, the niche of the brown bear is the one closest to that of the polar bear; the reconstructed node of their common ancestor is located precisely in the middle of both, in a similar way to what happens with the Asian black and the American black bears, with the difference that the latter present environmental overlap.

### 4.3. Geographic Ranges, Environmental Preferences, and Niche Trajectories

Species can have environmental niches that encompass both wide and restricted conditions, exhibiting contrasting and concordant niche trajectories and evolutionary patterns within and among species. For instance, although the American black bear is only distributed in North America, its phyloclimatic niche is notably broader than that of the brown bear, distributed across North America, Europe, and Asia. This is because the American black bear is adapted to a wider range of environmental conditions and currently includes regions where the brown bear has been locally extirpated [64]. On the other hand, the spectacled bear (*Tremarctos ornatus*) is the species more geographically isolated and the only one distributed exclusively in the Southern Hemisphere, living in the Andes Mountain range. Nonetheless, its phyloclimatic niche is not as restricted, showing a certain degree of environmental overlap with the American black bear and the sun bear (*Helarctos malayanus*). It also has a niche width like that of the American black bear, suggesting that, despite their distinct geographic ranges and evolutionary histories, both species share similar environmental preferences. They are adapted to comparable ecological conditions, exhibiting a considerable degree of ecological flexibility that allows them to inhabit a diverse range of habitats within their respective ranges, from dense forests to open plains. Contrastingly, the sun bear has a phyloclimatic niche encompassing the greatest range of precipitation conditions, but with a rather narrow tolerance to temperature conditions. This extensive variation in the precipitation axis could be related to the wide humidity requirements of the invertebrates included in this species diet, which, in many regions of its distribution, constitute its main food source [65].

The sloth bear (*Melursus ursinus*) is an example of a restricted phyloclimate niche, which shows the highest temperature conditions of all the species. This ecological characteristic is likely related to its specialized diet, which consists mainly of ants and termites [66]. The diversity of termites is highest in tropical and subtropical regions of the world, linked to metabolic processes like digestion and colony growth, which increase in warmer temperatures [67,68]. Interestingly, the distribution of the sloth bear is sympatric with the Asian black bear and the sun bear in southern Asia, sharing similar climatic conditions and having diverged from their common ancestor relatively recently, less than one million years ago.

Finally, the overlapping geographic distributions and shared phyloclimatic niches between the sun bear and the Asian black bear in certain regions of Asia suggest a scenario of competitive exclusion [69,70]. These species rely on similar food resources like fruits, insects, and vegetation found in the forest understory. This overlap in dietary preferences and habitat utilization can result in interspecific competition for food, space, and other critical resources. How they compete for resources can be associated with their differences in body size, behavior, foraging strategies, and feeding habits. The Asian black bear is larger in size and dominates in the most fruit-rich habitat, while the smaller sun bears predominantly consume insects; thus, the former has a competitive advantage in accessing certain resources or defending territories [69,70].

### 4.4. Implications for Conservation and Management

The insights gained from our integrated phylogeography and ecological niche modeling approach can have profound implications for the conservation and management of bear species, particularly in the context of ongoing climate change and habitat degradation. For instance, our findings of the phyloclimatic niches of the four *Ursus* species underscore the dynamic interplay between niche conservatism and niche evolution in shaping the evolutionary trajectories of bear species. It also highlights the importance of considering both factors for the interpretation of evolutionary history, as well as for conservation and management strategies. In general, some lineages maintain ancestral niche characteristics over evolutionary time, while others undergo niche shifts in response to environmental changes or ecological opportunities. Understanding these dynamics is crucial for elucidat-

ing the mechanisms driving species diversification and distribution patterns within the Ursidae family.

By characterizing the environmental niches and evolutionary trajectories of each bear species, we can better anticipate their responses to shifting climatic conditions and identify areas of conservation priority. We can emphasize three examples that concern the polar bear, the giant panda, and the sun bear. Considering that the polar bear phyloclimatic trajectory has led this species towards a region of extreme environmental and low diversity conditions, it is significantly exposed to negative effects of global climate change. Notably, the estimated ecological niche of the giant panda is the smallest of the entire family, and its geographic distribution is markedly shrunken by a myriad of anthropic impacts. Finally, the estimated niche of the sun bear encompasses the narrowest temperature range present in the entire bear family, leaving this species potentially more vulnerable to changes in this climatic variable.

Additionally, incorporating phylogenetic information into conservation planning can support the prioritization of evolutionarily distinct lineages and the preservation of evolutionary potential within bear populations. Furthermore, conservation efforts can be tailored by considering not only current distributions, but also historical environmental niches, to inform areas that are likely to support species' adaptation and persistence in the face of future environmental changes. In conclusion, our study allowed us to elucidate potential patterns of niche conservatism and evolution, providing valuable insights into the ecological and evolutionary processes driving the diversification and distribution of the Ursidae family. These insights are essential for guiding effective conservation strategies and ensuring the long-term survival of these iconic mammalian taxa.

**Supplementary Materials:** The following supporting information can be downloaded at: https://www.mdpi.com/article/10.3390/d16040223/s1, Sequence alignment: Mitogenome sequence alignment in phylip format.

**Author Contributions:** Conceptualization, C.L.-A. and E.V.-D.; formal analysis, C.L.-A.; investigation, C.L.-A. and E.V.-D.; methodology, C.L.-A. and E.V.-D.; resources, E.V.-D.; supervision, E.V.-D.; writing of the original draft, C.L.-A. and E.V.-D. All authors have read and agreed to the published version of the manuscript.

**Funding:** This research received no external funding.

**Institutional Review Board Statement:** Not applicable.

**Data Availability Statement:** All data used for the study are included in the manuscript and Supplementary Materials. No new code or scripts were developed.

**Acknowledgments:** C.L.-A. acknowledges that this paper was a part of his doctoral thesis in the Programa de Doctorado en Ciencias Biológicas de la Universidad Nacional Autónoma de México (UNAM). C.L.-A. had a scholarship and financial support provided by the Consejo Nacional de Ciencia y Tecnología (CONACyT CVU 508602/Reg. becario 288706). C.L.-A. was supported by a postdoctoral grant from Dirección General de Asuntos del Personal Académico (DGAPA, UNAM). This article was completed. while E.V.-D. was on sabbatical at the Estación Biológica de Doñana-CSIC with support from DGAPA, UNAM (PASPA No. 067/2023), and CONAHCyT.

**Conflicts of Interest:** The authors declare no conflicts of interest.

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
