# Peer review of "Bears into the Niche-Space: Phylogeography and Phyloclimatic Model of the Family Ursidae"

_diversity, doi:10.3390/d16040223_

Round 1

Reviewer 1 Report

Comments and Suggestions for Authors

This study focused on the eight extant bear species worldwide, investigating the issue of ecological niche evolution using phylogeography and phyloclimatic models along with ecological niche theory. Previous studies have mostly concentrated on phylogeography or used ecological niche theory and species distribution models to predict the suitable ecological niches of extant animals, with few studies combining the two to explore the impact of climate on species' evolutionary history. Therefore, this study is innovative and worthy of publication.

Regarding this study, I have three suggestions:

1. Among the eight species studied, there is a significant disparity in the number of distribution points, with the species Helarctos malayanus only having 35 points and Melursus ursinus 71 points. These data points for these two species are relatively low compared to other species with hundreds or even thousands of points. Does having 35 data points meet the minimum computational requirements of the model? Will the differences in data point quantities affect the results when compared to species with hundreds or thousands of points?

2. The database of occurrences used in this study were all sourced from GBIF. Could this approach overlook some useful species occurrences data information? While we acknowledge that GBIF is currently the largest and most comprehensive website for species distribution point data worldwide, it cannot guarantee the collection of distribution information for all species. I suggest that the authors consider collecting species distribution point information from published literature, especially non-English literature, to enrich the dataset and enhance the study.

3. The climate data used in this study are from the WorldClim database. However, when analyzing a species' evolutionary history, should historical climate data also be considered? It could be beneficial to compare historical climate data with modern climate data to assess the impact on this study.

Author Response

Included as pdf file

Reviewer 2 Report

Comments and Suggestions for Authors

In the paper “Bears into the niche-space: Phylogeography and phyloclimatic model of the family Ursidae” the authors evaluated their evolutionary patterns based on a combined phylogeography and niche modeling approach. This manuscript is well organized, and the drawn conclusions are coherent with the obtained results. Although I have loved reading your work, there are a few grammar issues that I have seen, thus I believe that the text needs to be edited by a native English speaker. I hope to provide very useful suggestions to improve the overall clarity of your study as well as the quality of your analysis. I think that my suggestions look feasible to you, and I believe you will be able to address them. Thus, please take care to do a full revision of your manuscript according to all my comments. Improvements based on my comments will be crucial for acceptance. I have some concerns and suggestions for each aspect of the manuscript. Please see below.

Lines 19 - 27: I would like to suggest giving more emphasis to the results.

Lines 28 – 29: Please arrange the keywords alphabetically.

Lines 31 – 106: The paper is technically sound and the claims are convincing. However I think that some references should be updated. Please, note that the hypothesis and the predictions are unclear, you need to well explain them.

Lines 35 - 37: I think that you should add these important references to support your sentence: “Ecological niche modeling is a set of techniques and specialized tools for studying the response of living beings to climatic variables that constitute their environment and their associated geographic distribution”. I would like to suggest:

Salinas-Ramos, V. B., et al., (2021). Artificial illumination influences niche segregation in bats. Environmental Pollution, 284, 117187.

Bosso, L., et al. (2024). Integrating citizen science and spatial ecology to inform management and conservation of the Italian seahorses. Ecological Informatics, 79, 102402.

Lines 75 – 86: Please, explain in detail your hypothesis and predictions. You need to expand this section if you would want to express exactly what you want to do.

Lines 108 – 203: Materials and methods: In general, the methods are appropriate and the study seems well conducted, although some details deserve a bit more attention i.e., especially about the methodology and the data. All the scripts/codes used in this paper must be added in the supplementary materials. Please, provide also all the link to source where you downloaded the data.

Lines 109 – 127: Please, add all the R code in the supplementary materials.

Line 123: Why did you choose to use layer at 10 arc-minutes?

Line 130: Please add the north symbol and the scale in the figure.

Lines 136 – 166: Please, add all the R code in the supplementary materials.

Lines 204 – 301: Well written! The figures and the tables are all informative and necessary, but not redundant, ensuring the correct comprehension of the manuscript.

Lines 304 – 460: The paper discussed appropriately the context and the theme, although there is important literature not cited by the authors. I think that the authors should be discussing their results also comparing them with those already published on other species/genus/family. In fact your paper discusses findings in relation to some of the work in the field but ignores other important work that I think should be added in your discussion considering also other factor of niche analysis.

Lines 343 – 344: I think that you should add this important reference to support your sentence: “However, it is noteworthy that the Asian black bear, the more ancient lineage, is better adapted to a wider variety of wet and warm climatic conditions.”. I would like to suggest:

Ahmad, F., et al., (2022). Patterns of spatial distribution, diel activity and human-bear conflict of Ursus thibetanus in the Hindu Kush mountains, Pakistan. Global Ecology and Conservation, 37, e02145.

Goursi, U.H., et al., (2021). Spatial distribution of the threatened Asiatic black bear in northern Pakistan. Ursus, 2021(32e13), 1-5.

Comments on the Quality of English Language

Moderate editing of English language required.

Author Response

Submitted as a pdf file

Reviewer 3 Report

Comments and Suggestions for Authors

Author Response

Included as a pdf file

Round 2

Reviewer 2 Report

Comments and Suggestions for Authors

Well done!

Comments on the Quality of English Language

Although I have loved reading your work, there are a few grammar issues that I have seen, thus I believe that the text needs to be edited by a native English speaker.